# Race-Specific Impact of Telehealth Advance Care Planning on Cost of Dementia: A Cost Prediction Study

**DOI:** 10.3390/ijerph22091399

**Published:** 2025-09-07

**Authors:** Peter S. Reed, Yonsu Kim, Jay J. Shen, Sai Kosaraju, Mingon Kang, Jennifer Carson, Iulia Ioanitoaia Chaudhry, Sarah Kim, Connor Jeong, Yena Hwang, Ji Won Yoo

**Affiliations:** 1Sanford Center for Aging, School of Medicine, University of Nevada, Reno, Reno, NV 89557, USA; 2Department of Health Behavior, Policy and Administration Sciences, School of Public Health, University of Nevada, Reno, Reno, NV 89557, USA; jennifercarson@unr.edu; 3Department of Healthcare Administration and Policy, School of Public Health, University of Nevada, Las Vegas, NV 89154, USA; yonsu.kim@unlv.edu (Y.K.); jay.shen@unlv.edu (J.J.S.); 4Center of Health Disparities Research, School of Public Health, University of Nevada, Las Vegas, NV 89154, USA; 5Department of Computer Science, California State Polytechnic University, Pomona, CA 91768, USA; skosaraju@cpp.edu; 6Department of Computer Science, Howard Hughes College of Engineering, University of Nevada, Las Vegas, NV 89154, USA; mingon.kang@unlv.edu; 7Dementia Engagement, Education, and Research (DEER) Program, School of Public Health, University of Nevada, Reno, Reno, NV 89557, USA; 8Veterans Affairs Southern Nevada Healthcare System, North Las Vegas, NV 89086, USA; iulia.ioanitoaia-chaudhry@va.gov; 9The Connection Sphere, Las Vegas, NV 89144, USA; sarahkimm13@gmail.com (S.K.); yenahwang16@gmail.com (Y.H.); 10Department of Internal Medicine, Kirk Kerkorian School of Medicine, University of Nevada, Las Vegas, NV 89154, USA; connorjeong@gmail.com (C.J.); ji.yoo@unlv.edu (J.W.Y.)

**Keywords:** cost, health disparity, dementia, advance care planning, workforce education

## Abstract

Identifying strategies to enhance patient engagement and to control healthcare costs promotes a responsive and efficient healthcare system. The aim of this study is to predict healthcare cost savings associated with delivering telehealth advance care planning (ACP) to patients living with dementia. Two Geriatrics Workforce Enhancement Programs delivered training to primary care providers on using telehealth to provide ACP. Using electronic health records data from 6344 dual-eligible Medicare/Medicaid patients receiving telehealth primary care from trained providers in an urban safety net system, persons living with dementia (*n* = 401) were identified by extracting ICD-10 codes. The primary outcome was the estimated hospitalization-associated cost, with a key independent variable of ACP billing status. Multiple linear regressions and machine learning techniques estimated the impact of telehealth ACP on hospitalization-associated costs with a differential analysis by race. Compared to non-Hispanic Whites, hospitalization costs among Hispanic elders were higher by USD 14,232.40. Costs for non-English speakers or those having increased comorbidities were higher by USD 27,346.60 and USD 26,072.70, respectively. Overall, receiving ACP was associated with lower costs of USD 23,928.84. Dementia patients seen by primary care providers in a system receiving training to offer ACP via telehealth realized significant cost savings, with marked differences among those of non-White racial backgrounds.

## 1. Introduction

Approximately seven million people are estimated to be living with Alzheimer’s disease or other causes of dementia (referred to as ‘dementia’ in this paper) in the United States [1]. Due to a range of systemic challenges regarding early diagnosis and lack of treatment options, along with various fragmentations in the healthcare system, the responsibility for dementia care is often transferred to family care partners and community stakeholders. Further, there are disparities across various groups in the effectiveness of caring for persons living with dementia [2]. These disparities are more pronounced in economically disadvantaged populations, including those living with dementia who are covered by both Medicare and Medicaid insurance (i.e., dual eligible) [2]. The average annual Medicaid payments per person for dual-eligible Medicare beneficiaries with dementia in 2022 (USD 6739) were 22 times higher than the payments for those without dementia (USD 303) [2].

Alongside these costs, the state of Nevada is a designated healthcare provider shortage area [3], a challenge that is worsening over time with rapid population growth. This enormous healthcare demand–supply gap results in lower effectiveness and efficiency of healthcare for persons living with dementia. In 2022, Nevada ranked first in Medicare expenditure (USD 8426 per capita) due to emergency department visits and hospital admissions, and it is projected to experience the second-highest increase in Medicaid payments by 2025 (36.5%), behind only Alaska (44.6%) [2].

After the Centers for Medicare and Medicaid Services (CMS) expanded telehealth coverage in the U.S. during the public health emergency response to the COVID-19 pandemic, one in ten Medicare beneficiaries accessed telehealth services [4]. With this expansion, it is essential to ensure consistent quality of care when using telehealth. One opportunity to support the quality of telehealth primary care services is through more widespread adoption of the Age-Friendly Health System 4M framework [5]. More consistently addressing the ‘4Ms’ of high-quality care for older adults would involve an increasing focus, in all patient interactions, on the following elements: What ‘Matters’ to the patient, ‘Medication’, ‘Mentation’, and ‘Mobility’ [5].

In supporting the concept of what matters to the patient, advance care planning (ACP) documentation, along with other discussions with patients to identify key priorities and preferences, has been recommended to occur within two years of a dementia diagnosis [6]. ACP plays a critical role in aligning care plans from healthcare provider teams to increase efficiency when serving persons living with dementia across care settings [7,8,9,10]. Although CMS reimburses primary care providers for engaging patients in discussions about their care preferences, ACP has been underutilized for Medicare beneficiaries [11]. Telehealth may facilitate ACP discussion and documentation when access to healthcare is limited, including supporting patients in rural and underserved communities or during a crisis (such as the COVID-19 pandemic) [12]. However, telehealth itself may not be a panacea for access, as disparities in telehealth access (and quality) have been observed among racial and ethnic minority older adults [4,12,13].

To respond to these challenges, two United States Health Resources and Services Administration-funded Geriatrics Workforce Enhancement Programs (GWEPs) in Nevada, along with Comagine Health, the Nevada Aging and Disability Services Division, and other partners, collectively launched the Nevada Geriatrics Telehealth Collaborative (NGTC). The NGTC was part of a larger effort known as the Nevada COVID-19 Aging Services Network Rapid Response (or Nevada CAN), which was launched during the pandemic to promote improved access and quality of telehealth services for the state’s older adults, including persons living with dementia and their care partners. The NGTC included a focus on offering innovative training for primary care providers on how to incorporate patient-centered interventions into telehealth services, with a focus on key competencies aligned with the 4Ms of an Age-Friendly Health System framework [13,14,15,16].

In this study, we aimed to examine telehealth ACP’s race-specific impact on hospitalization-associated costs among persons living with dementia, using novel machine learning techniques to assess outcomes in a health system where providers had access to geriatrics training. Machine learning has been recognized for demonstrating better performance in predicting healthcare expenditures than traditional analytical approaches [17]. Further, mitigating race and ethnicity bias in healthcare has also been implemented by using machine learning fairness algorithms [18]. Thus, to better assess the racial- and ethnic-specific impact on the ACP’s cost-saving role, machine learning models offer a useful analytical tool. Bringing all these elements together, this study provides insight into future strategies for leveraging telehealth ACP to support diverse patients diagnosed with dementia in contributing to their own care planning to support equitable cost savings.

## 2. Materials and Methods

In this complex study, leveraging machine learning models to create a cost prediction for healthcare utilization among people living with dementia who have (and have not) received telehealth ACP, there are a range of important methodological issues to highlight. These are broken down to provide key details in the following subsections: (1) Study design and sample overview; (2) Outcome and data sources; and (3) Statistical methods and analyses.

### 2.1. Study Design and Sample Overview

This was a retrospective, cross-sectional design study of dual-enrolled Medicare- and Medicaid-insured community-dwelling patients diagnosed with dementia who received telehealth services from an urban, academic, not-for-profit ambulatory clinic system in Nevada. Study participant selection was conducted in accordance with the Strengthening the Reporting of Observational Studies in Epidemiology (STROBE) statement [19]. Clinic electronic health records (EHRs) of 6344 patients with at least one video telehealth visit between 1 January 2022 and 31 December 2022 in Nevada were analyzed. Telehealth visits were provided for longer than 15 min. by licensed primary care providers within a system offering provider training on delivering Age-Friendly Health System services to address What Matters, Medication, Mobility, and Mentation. Patients (5847) without a diagnosis of dementia (International Classification of Diseases, 10th edition, Clinical Modification (ICD-10-CM) codes of either F01, F02, or F03) were excluded from the study. Among 497 patients with a dementia diagnosis identified, 96 were excluded due to incomplete data. Ultimately, a total of 401 patients were selected for final analysis.

### 2.2. Outcome and Data Sources

The primary study outcome was hospitalization-associated healthcare costs for 401 patients with a diagnosis of dementia. Hospitalization-associated information was collected from hospital discharge summaries, including the principal diagnoses and hospital length of stay information from the EHRs. The verification of this information was supported by the Medicare and Medicaid payor claims data. The hospitalization-associated healthcare costs were estimated from the Nevada State Inpatient Database (SID) between 1 January 2021 and 31 December 2021. The Nevada SID contains hospital discharge records of all community hospitals in the state of Nevada, developed for the Healthcare Cost and Utilization Project by the Agency for Healthcare Research and Quality [20]. The Nevada SID files were constructed from hospital discharge files received from the UNLV Center for Health Information Analysis under the authority of the Nevada Division of Healthcare Financing and Policy [20]. The Nevada SID data includes de-identified patient-level demographics, diagnostic and procedure codes, and age- and gender-adjusted hospital charges by principal diagnoses.

Using these data, the specific hospitalization-associated cost was estimated by integrating the hospital length of stay and the daily hospital charges. According to CMS’s estimated hospital care annual growth rate in 2022, 0.8% was applied to estimate 2022 costs from the SID 2021 hospitalization-associated costs [21]. Additional patient information was collected from the clinic EHRs to support the overall analysis of the key variables of interest, including age, gender, race and ethnicity, preferred language, ACP billing, and the Charlson Comorbidity Index (CCI). ACP billing documentation was defined as the presence of Current Procedural Terminology (CPT) codes 99497 or 99498. Race and ethnicity information was self-reported. For this analysis, race was divided into non-Hispanic White, Black, Hispanic; and Asian/Hawaiian/Pacific Islander (AHPI). The preferred language information was self-reported and divided into English and other languages. The CCI was obtained as a comorbidity measurement [22].

### 2.3. Statistical Methods and Analyses

A chi-square test for categorical variables (gender, preferred language, ACP) and an analysis of variance (ANOVA) test for continuous variables (age, CCI, cost) were applied to examine the statistical significance of race groups. To test our hypothesis, we estimated multiple linear regressions with cost as the dependent variable. The predictors included age, gender, race, preferred language, ACP, and CCI. We obtained the standard error, coefficient estimate, 95% confidence intervals, and *p*-value of each predictor. To avoid multi-collinearity, the Variance Inflation Factor (VIF) was measured [23]. A stepwise approach was used to obtain analyses from the parsimonious regression models [23]. An adjusted R-squared was obtained to evaluate the proportion of the variance in the response variable that can be explained by the predictors in the model. All statistical analyses were two-tailed, and a *p*-value < 0.05 was considered statistically significant. SAS version 9.4 (Cary, NC, USA) was used for the statistical analysis.

We compared the cost prediction performance across four racial groups using six supervised machine learning regression models: linear regression (LR), support vector regression (SVR), decision tree regression (DTR), Lasso regression (LassoR), random forest regression (RFR), and neural network (NN). The input feature set comprised both categorical and continuous variables. The categorical features included gender, preferred language, and ACP status, each encoded as binary variables, while the continuous features included age and CCI. The target label for model training and evaluation was the cost variable, a continuous numerical value. Let X ∈ ℝ^(n×d)^ denote the complete feature matrix and y ∈ ℝ^n^ denote the cost values for *n* samples. Continuous features were standardized using z-score normalization, as shown below.(1)$$x_{ij}^{\prime }=\frac{\left(x_{ij}-\mu_{j}\right)}{\sigma_{j}}$$
where μ_j_ and σ_j_ represent the mean and standard deviation of feature j in the training set, with these parameters applied to normalize the test set, and the models were trained using stratified 5-fold cross-validation to ensure proportional representation of racial groups in each fold. Hyperparameters were tuned via grid search with an 80–20% split of each training fold into training and validation subsets. The selected hyperparameters included an L2-regularization parameter of λ = 0.8 for LR, an SVR model with a radial basis function (RBF) kernel and L2-norm regularization parameter C = 0.8, and an NN with a single hidden layer of eight nodes using the rectified linear unit activation function given below.(2)$$ReLU(z)=max\left(0,z\right)$$

For optimization, the LR/LassoR objective minimized the mean squared error with an $L_{p}$_p penalty term, as follows.(3)$$\underset{w,b}{\min}\frac{1}{n}\sum_{i=1}^{n}{\left(y_{i}-w^{T}x_{i}-b\right)}^{2}+\lambda {\left|\left|w\right|\right|}_{p}$$

The SVR objective minimized margin loss subject to ε-insensitive constraints, as shown below.(4)$$\underset{w,b}{\min}\frac{1}{2}{\left|\left|w\right|\right|}^{2}+C\sum_{i=1}^{n}\left(\xi_{i}+\xi_{i}^{*}\right)$$

Subject to(5)$$y_{i}-w^{T}x_{i}-b\;\leq \;\epsilon +\xi_{i},\;w^{T}x_{i}+b-y_{i}\;\leq \;\epsilon +\xi_{i}^{*},\;\xi_{i},\xi_{i}^{*}\;\geq 0.\;\;$$

Model performance was evaluated using the root mean square error (*RMSE*), computed separately for each racial group in each fold, as defined below.(6)$$RMSE=\sqrt{\frac{1}{n}\sum_{i=1}^{n}{\left(y_{i}-\hat{y_{i}}\right)}^{2}}$$

The experiments were repeated 20 times for reproducibility. Statistical differences in *RMSE* distributions among racial groups were assessed using the Wilcoxon rank-sum test, a non-parametric method that does not assume normality.

This study received ethical approval from the University of Nevada, Las Vegas Institutional Review Board (IRB #1510973-5, overall; IRB #2024-18, ADRD).

## 3. Results

Of the 401 participants, described in Table 1, 50.3% were non-Hispanic White, 19.5% were Black, 17.8% were Hispanic, and 12.4% were Asian/Hawaiian/Pacific Islander (AHPI). The mean age was 74.8 years, with a standard deviation (SD) of 6.5 years and a range of 65–92 years. Racial minorities were younger, on average, than non-Hispanic White participants. Females accounted for 54.0%. In terms of the preferred language, 33.8% preferred languages other than English, and racial minorities preferred non-English languages more than non-Hispanic Whites. In terms of the CCI, the mean CCI was 3.9, with an SD of 1.5, a minimum of 2, and a maximum of 8. Compared with non-Hispanic Whites, racial minorities had higher CCIs. Additionally, 26.5% were documented for the ACP billing. Compared with non-Hispanic Whites, racial minorities had lower rates of ACP billing. In terms of cost, the mean cost was USD 52,401.30, with an SD of USD 97,644. Compared with non-Hispanic Whites, racial minorities had higher costs (See Table 1).

Table 2 demonstrates the regressions of cost. Older age was marginally associated with higher costs by USD 1331.10 [95% CI USD −76.5, USD 2738.5] (*p* = 0.06). Compared with those of non-Hispanic Whites, the costs of healthcare for Hispanics were higher by USD 14,232.40 [95% CI USD 3033.80, USD 25,431.00] (*p* = 0.01). Compared with those with English as the preferred language, the costs of those with other preferred languages were higher by USD 27,346.60 [95% CI, USD 5996.10, USD 48,697.10] (*p* = 0.01). Higher comorbidities were associated with higher costs by USD 26,072.70 [95% CI USD 19,780.30, USD 32,365.10] (*p* < 0.001). Compared to those without ACP, receiving ACP was associated with reduced costs averaging USD −23,928.84 less [95% CI USD −43,232.20, USD −4625.40] (*p* = 0.01). Females or those with Black or AHPI race did not demonstrate higher costs compared with males or non-Hispanic Whites, respectively. The adjusted R-squared value for the model was 0.24. There was no multicollinearity concern, as all the VIFs were less than 1.5. Figure 1 presents the cost predictions by race.

Through machine learning models, the lowest *RMSE* was found among non-Hispanic Whites, followed by Black, Hispanic, and AHPI individuals. The lowest *RMSE* was statistically significant compared with the other racial groups (Wilcoxon rank-sum *p* < 0.01). This implies that machine learning models for cost prediction are biased toward non-Hispanic Whites. The bias is mainly explained by the imbalance in data size. In the authors’ judgment, among the machine learning models used in this study, NN demonstrated the best performance. NN captures non-linear associations between the data extracted from the EHR and the cost.

## 4. Discussion

We aimed to better understand the impact of advance care planning (ACP) delivered via telehealth on hospitalization-associated costs and differences by race among patients diagnosed with dementia. To the best of our knowledge, this is the first report confirming ACP’s cost-saving role for community-dwelling persons living with dementia that includes examining its role from a health equity perspective. The results demonstrated that ACP billing was associated with fewer hospitalization-associated costs. We believe providing telehealth ACP counseling helped overcome delayed access to healthcare. Our findings also lend support for potential policies regarding Medicare beneficiaries, highlighting potential payment reforms to achieve greater equity and higher values in caring for persons living with dementia [24].

As we analyzed Medicare and Medicaid dual-eligible participants in primary care provider shortage areas, our findings note disparities in access to patient-centered ACP and call for efficient healthcare innovation for persons living with dementia. Further, we believe that the benefits of enhancing healthcare efficiency have the potential to enhance the experience and health of unpaid family care partners of persons living with dementia. Timely and optimal care-partner training and referral to community resources, including respite care, are aligned in the process of establishing ACP. Through identifying potential community-based supports and services to complement primary care and by outlining those as recommendations in a care plan, opportunities for increasing support are developed.

As a result, the cost of hospitalization, and potentially other healthcare-related costs, can be reduced, particularly when the available supports enable the persons living with dementia to remain in their homes and communities. These coordinated care cycles from initiating the ACP discussion are particularly beneficial for socially disadvantaged populations, rural area residents, and racial and ethnic minorities who have limited access to health and long-term care. The ACPs’ spillover effects, supporting community-based aging in place, are essential given the trend of declining nursing home beds by almost 25% over the past decade across the United States [25].

Our findings support deploying a learning health system that shares important end-of-life wishes and health data for persons living with dementia in bidirectional communications with healthcare providers, care partners, and community stakeholders [7,26]. In the Nevada Physician Order Life-Sustaining Treatment (POLST), a patient’s wishes are incorporated into a healthcare provider’s orders, which are recorded on a unique and brightly colored form that is kept in the front of the medical record or with the patient [27]. The Nevada Lockbox learning health system in Nevada shares documented ACP with care partners, authorized healthcare providers, and legal experts in a secure blockchain environment [28].

To the best of our knowledge, the present study is the first machine learning cost prediction study to evaluate race-specific impacts related to telehealth-based ACP discussions. It is worthwhile to note that as machine learning and artificial intelligence are adopted in healthcare, particularly healthcare for persons living with dementia, there may be opportunities to better understand and mitigate racial and ethnic disparities in outcomes [9,18]. Our findings of machine learning techniques might enhance the integrity of translational research from real-world practice to systems-based clinical quality improvement and population-level policy suggestions. The deeply rooted health disparities present within healthcare in the United States have negative effects on health outcomes and public health costs that can be magnified when race/ethnicity and age intersect [29].

Enhancing the delivery of ACP for racial and ethnic minorities and those preferring languages other than English during telehealth-delivered services may improve ACP’s cost savings. We found low rates of ACP billing documentation in racial minorities compared with the non-Hispanic White populations, particularly among Hispanic participants, with a rate of preferring a language other than English almost three times higher. From these findings, we highlight the priorities of cultural and linguistic sensitivities in ACP geriatrics workforce education for the racial and ethnic minority representation populations, as in the current study [30]. Our findings also highlight the improvement opportunities for Medicaid enrollees living with dementia, especially for those at risk of transition from homes and communities to institutions [31].

### Study Limitations

The findings of this study are limited by its exclusive focus on dual-eligible Medicare/Medicaid patients in a single health system in a Southwestern urban area, as well as by the one-year observation period. Further studies with different geographic populations, organizational/payor programs, and multiple years of observation might yield different results. For the administrative data extraction and review without including personally identifying information, we did not review the progress notes of healthcare providers in the EHRs to determine the degree of dementia.

Another limitation of the study was the self-reported racial information that was collapsed into three kinds of races and two ethnicities, and which did not include all races or mixed ones. The hospitalization-associated cost was estimated from the charge rates of the health system, which did not share actual reimbursement rates. We did not include data on healthcare-supporting systems like home health or community resources in the present study; therefore, factors determining access to telehealth care for persons living with dementia have not been adjusted. Thus, our analysis could be considered preliminary until further, more representative data are analyzed to confirm our findings.

Overall, this study finds that primary care providers with access to the 4Ms training offered by the NGTC who provided ACP via telehealth to patients diagnosed with dementia helped reduce hospitalization costs in a provider shortage area. Further, these findings were more pronounced among patients from racial minority groups.

## 5. Conclusions

Overall, this study demonstrated that primary care providers with access to 4Ms-based geriatrics training who support their patients with advance care planning (ACP) guidance during telehealth patient visits reduced hospitalization costs in a provider shortage area among patients diagnosed with dementia. These findings were even more pronounced among patients from racial minority groups, with increased reductions in costs among the non-White population. These results fulfilled our study aim to examine telehealth ACP’s race-specific impact on hospitalization-associated costs among persons living with dementia, using novel machine learning techniques to assess outcomes in a health system where providers had access to geriatrics training. Specifically, dementia patients seen by primary care providers in a system receiving training to offer ACP via telehealth realized significant cost-saving benefits, with marked differences among those of non-White racial backgrounds. These findings reinforce the importance of developing and delivering high-quality geriatrics training to primary care providers, including content encouraging providers to engage in discussions about care preferences and care planning as a means to directly reduce the costs of healthcare. The implications for the larger United States healthcare system in supporting dual-eligible (Medicare/Medicaid) patients, particularly among patients from non-White racial groups, demonstrate opportunities for expanded dementia and ACP education as a means of reducing costs.

## Figures and Tables

**Figure 1 ijerph-22-01399-f001:**
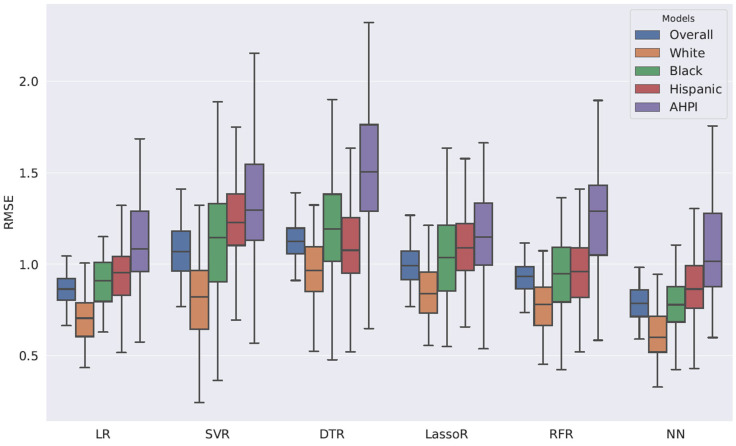
Root Mean Square Error Distributions for Various Machine Learning Regression Models.

**Table 1 ijerph-22-01399-t001:** Descriptive statistics by race.

		Race				
Variables	Total *n* = 401	Non-Hispanic White, % (*n*)	Black, % (*n*)	Hispanic, % (*n*)	AHPI, % (*n*)	*p*-Value *
	% (*n*)	50.3% (207)	19.5% (80)	17.8% (73)	12.4% (51)	
Age, mean (SD)	74.8 (6.5)	73.4 (7.3)	76.7 (5.6)	76.3 (5.8)	75.2 (6.5)	<0.001
Female	54.0% (222)	51.2% (106)	56.3% (45)	56.2% (41)	58.8% (30)	0.75
Preferred other than English	33.8% (139)	19.8% (41)	38.7% (31)	56.2% (41)	51.0% (26)	<0.001
Charlson Comorbidity Index, mean (SD)	3.9 (1.5)	3.5 (1.4)	4.0 (1.4)	4.1 (1.5)	4.0 (1.3)	<0.001
Advance care planning	26.5% (109)	32.4% (67)	22.5% (18)	19.2% (14)	19.6% (10)	0.013
Cost, mean ** (SD)	$52,401.3 ($97,644.3)	$33,555.0 ($82,359.1)	$67,452.7 ($106,392.0)	$70,718.2 ($102,365.1)	$67,679.7 ($178,763.9)	<0.001

* *p*-values for ANOVA test. ** 2022 USD. Abbreviations: AHPI, Asian Hawaiian Pacific Islander; SD, standard deviation.

**Table 2 ijerph-22-01399-t002:** Multiple linear regression of cost results.

				95% Confidence Interval		
Predictors	Referent	Standard Error	Coefficient Estimate	Lower	Upper	*p*-Value
Age	65	715.9	1331.1	−76.5	2738.5	0.06
Female	Male	8598.3	−10,361.6	−27,068.3	6345.2	0.22
Black	Non-Hispanic White	11,351.8	15,317.9	−7039.5	37,675.4	0.17
Hispanic	Non-Hispanic White	5685.2	14,232.4	3033.8	25,431.0	0.01
AHPI	Non-Hispanic White	13,656.2	19,786.1	−7126.9	46,699.2	0.14
Other than English	English	10,857.6	27,346.6	5996.1	48,697.1	0.01
Charlson Comorbidity Index		3200.8	26,072.7	19,780.3	32,365.1	<0.001
Advance care planning	Yes	9819.1	−23,928.84	−43,232.2	−4625.4	0.01

Adjusted R-squared = 0.24. Abbreviation: AHPI, Asian Hawaiian Pacific Islander. Cost values are in 2022 US dollars.

## Data Availability

To protect private patient health information, the data presented in this study are available on request from the corresponding author.

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
