# Peer review of "Race-Specific Impact of Telehealth Advance Care Planning on Cost of Dementia: A Cost Prediction Study"

_ijerph, 2025, doi:10.3390/ijerph22091399_

Round 1
Reviewer 1 Report
Comments and Suggestions for Authors
The paper addresses a machine learning study for cost prediction for telehealth advanced care planning for patients with dementia. The topic is interesting and falls within the scope of the journal. The aim of the work is to examine telehealth ACP's race-specific impact on hospitalization-associated costs among patients with dementia, using machine learning techniques to assess outcomes in a health system.
The comments are divided into major and minor comments and the overall merit.
Starting with the major points:
-Starting the Abstract with "in 2022", and we are in 2025, you need to rephrase the 1st sentance to best introduce the title which is mainly focusing on , advanced care, telehealth, machine learning, cost prediction, dementia.
-The population number 401 is too small for studies employing prediction using machine learning.
-In the introduction section lines 100 to 103: the objective was stated, however after the objective it is expected to see short summary on the approach followed in addition to the impact of such approach on the healthcare system, public health etc.
-There has to be a link between the paragraphs, for instance, a link between the end of introduction to the section of materials and methods and so on so forth.
-Section 2 "Materials and Methods" was not introduced. Before going to subsections directly, it is recommended to write a paragraph introducing what is going to be included in this section, before beginning with the subsections.
-It was expected to start section 2 with both a paragraph and a block diagram or (smart art) showing the steps of your methodology and materials from the input to the output, as a road map for the work, (this figure is significant).
-Add the images of the materials used, anonimous reports of patients, processing program interfaces etc..
-Again line 119 "411 patients were selected for final analysis. " the number is too small for machine learning applications and for carrying out solid conclusions. Noteworthy that in the abstract it was 401, here it is 411, what is the difference ?
-Add equations in the "Statistical Analysis method" with references.
-Before and After the "statistical evaluation method", nothing was specified on the machine learning classifiers, features, training methods, ANOVA etc.. the used methods under theses keywords have to be described in details.
-Also in the "data collection" subsection, the percentage of training and testing was not provided, please provide these details.
-There has to be a subsection on "ethical considerations" after the data collection because the study is dealing with patients, also the IRB can be added in this subsection.
-Results started directly with "Of the 411 participants, described in Table 1, 50.3% were non-Hispanic White, 19.5% were Black, 17.8% were Hispanic etc." It is recommended to link this paragraph with what was mentioned before, also to describe the parameters and demographics reported in Table 1 then reading and analyzing what is specified in the Table.
-In the "result" section these names appeared : AHPI, Asian Hawaiian Pacific Islander; DTR, decision tree regression; LassoR, Lasso regression; LR, linear regression; NN, neural network; RFR, random forest regression; RMSE, root mean square errors; SVR, support vector machine regression.
without any prior description of these networks/metrics in the methodology section.
-In the "discussion" section, add a table to show the results of the error and other metrics to existing papers in the literature review (comparative table).
-In the conclusion it was mentioned that NN showed the best result, there has to be a justification based on what ? what was the reference ? and what are the charactersitics and architecture of this best NN ?
-The conclusion can be more elaborated to link dimentia, to Machine learning to training programs to telehealth to the cost.
-Section of Limitations of the work can be added.
Minor points :
-Why the M is in bold lines 71 to 72 ?
- Line 120 I suggest "Resources" instead of "Outcome and measures", recall that this subsection is under the methodology section rather than the result section.
-Line 148 I suggest "Statistical analysis Methods" to emphasize that you are still in the method section.
-In the reference section, more articles can be added and cited.
Overall Merit :
It is recommended to present more results to show the novelty of the work and compare it to peers, also to describe in details the methodology and materials while increasing the population number when using ML for prediction.
Author Response
Please see attachment for responses to all reviewers' comments.

Reviewer 2 Report
Comments and Suggestions for Authors
The study on "Race-specific impact of telehealth advance care planning on cost of dementia: A machine learning cost prediction" is significant considering that the prevalence of Dementia is on the rise worldwide especially in the elderly. Therefore, the telehealth advance care would be beneficial specially in developing countries where the healthcare cost is unaffordable.
How ever these are my comments to the Authors:
- The abstract lacks the order into Background, Aim, Methods, Results, convulsions
- The authors should try to adjust the break-words for example on lines 22, 23, 24 and 31.
- The word "care" should be completed as "healthcare" throughout the manuscript.
- All abbreviations should be listed at the end of the document as a few have been listed.
- In discussions of findings, there should be supported with similar studies performed elsewhere. This is the essence of performing a literature review to fill in the gaps of other studies.
- In the data analysis, the authors stated that associations were performed but nothing is reported in the abstract.
- In the 1st LONG paragraph of the INTRODUCTION reference [2] is monotonous. Other references should be stated.
- Throughout the study the LONG paragraphs should be divided into small paragraphs for readers to be able to read what has been stated.
- There should be consistency in the writing of references e.g references 14, 15, 16, and 28.
- Old references below years 2020 should be replaced with recent ones.
Reviewer 3 Report
Comments and Suggestions for Authors
WELL WRITTEN ARTICLE INFORMATIVE relevant . Authors may consider whether " preliminary report " as mentioned by them in the limitations can be added in the title. Are the words " Machine learning' necessary in the title -- the emphasis should be on the observations and inferences rather than on using ML - which in any case is not described in great detail . Some of the sentences could be briefer and repetition reduced.
Comments on the Quality of English LanguageSome of the sentences could be briefer and repetition reduced.
Round 2
Reviewer 2 Report
Comments and Suggestions for Authors
Dear Authors,
My responses to former review and more comments are attached below.
Thanks for the great work. The reviewed article is much better. I am sorry for the delays and inconveniences that could have been caused.
Kind regards,
